

# Examining cross-scale influences of forcing resolutions in a hillslope-resolving, integrated hydrologic model

Miguel A. Aguayo[1], Alejandro N. Flores[2], James P. McNamara[2], Hans-Peter Marshall[2], and Jodi Mead[2]

[1]Universidad Católica de Temuco, Montt 56, Temuco, Chile
[2]Boise State University, 190 University Drive, Boise, ID 83725

**Correspondence:** Miguel A. Aguayo (miguel.aguayo@uct.cl)

**Abstract.** Water management in semiarid regions of the western United States requires accurate and timely knowledge of runoff generated by snowmelt. This information is used to plan reservoir releases for downstream users and hydrologic models play an important role in estimating the volume of snow stored in mountain watersheds that serve as source waters for downstream reservoirs. Physically based, integrated hydrologic models are used to develop spatiotemporally dynamic estimates of hydrologic states and fluxes based on understanding of the underlying biophysics of hydrologic response. Yet this class of models are associated with many issues that give rise to significant uncertainties in key hydrologic variables of interest like snow water storage and streamflow. Underlying sources of uncertainty include difficulties in parameterizing processes associated with nonlinearities of some processes, as well as from the large variability in the characteristic spatial and temporal scale of atmospheric forcing and land-surface water and energy balance and groundwater processes. Scale issues, in particular, can introduce systematic biases in integrated atmospheric and hydrologic modeling. Reconciling these discrepancies while maintaining computational tractability remains a fundamental challenge in integrated hydrologic modeling. Here we investigate the hydrologic impact of discrepancies between distributed meteorological forcing data exhibiting a range of spatial scales consistent with a variety of numerical weather prediction models when used to force an integrated hydrologic model associated with a corresponding range of spatial resolutions characteristic of distributed hydrologic modeling. To achieve this, we design and conduct a total of twelve numerical modeling experiments that seek to quantify the impact of applied resolution of atmospheric forcings on simulated hillslope-scale hydrologic state variables. The experiments are arranged in such way to assess the impact of four different atmospheric forcing resolutions (i.e., interpolated 30 m, 1 km, 3 km and 9 km) on two hydrologic variables, snow water equivalent and soil water storage, arranged in three hydrologic spatial resolution (i.e., 30 m, 90 m and 250 m). Results show spatial patterns in snow water equivalent driven by atmospheric forcing in hillslope-scale simulations and patterns mostly driven by topographical characteristics (i.e., slope and aspect) on coarser simulations. Similar patterns are observed in soil water storage however, in addition to that, large errors are encountered primarily in riparian areas of the watershed on coarser simulations. The Weather Research Forecasting (WRF) model is used to develop the environmental forcing variables required as input to the integrated hydrologic model. WRF is an open source, community supported coupled land-atmosphere model capable of capturing spatial scales that permit convection. The integrated hydrologic modeling framework used in this work coincides with the ParFlow open-source surface-subsurface hydrology model. This work has important implications for the use of atmospheric and integrated hydrologic models in remote and ungauged areas. In particular, this work has





potential ramifications for the design and development of observing system simulation experiments (OSSEs) in complex and snow-dominated landscapes. OSSEs are critical in constraining the performance characteristics of Earth-observing satellites.

# 1 Introduction

In semi-arid mountain areas of the western United States, snow accumulation and melt are the most critical factors to generate sufficient water supply for human and ecological systems. Water resources management in these areas requires accurate and timely knowledge of runoff generation by snowmelt (Dettinger, 2005; Stewart et al., 2005). Generally, those management strategies are based on the use of lumped hydrologic models (Burnash et al., 1973; Brunner, 2010; Havnø et al., 1995; Mishra and Singh, 2002) because of their simplicity in data needs, historically proven performance, and computational efficiency.

The parameterized nature of key watershed processes and properties in these models, however, necessitates data records of significant length for calibration and cannot respond to disturbances internal to the watershed such as land use change, or changes in vegetative cover associated with climate change or other drivers. Because these models require sufficiently long periods of observed precipitation and streamflow it is also impossible to validate predictions in ungauged basins.

Integrated hydrologic models, on the other hand (Abbott et al., 1986; Butts et al., 2004; Kollet and Maxwell, 2006; Inc, 2015),
are more sophisticated in their representation of processes and, in principle, provide predictions of observable parameters (e.g., snow water equivalent, soil moisture, etc.) throughout the watershed rather than only at the watershed outlet. Even though physically based models represent most hydrologic processes with more fidelity to hydrologic processes than lumped parameter models, they suffer from a number of other issues that has made their general use for hydrologic forecasting difficult. First, nonlinearities and closure problems in the underlying processes ultimately necessitate empirical parameterization (e.g.,
constitutive relationships between soil water content and matric potential). Second, these models require a correspondingly complex and large amount of spatiotemporally varying data related to atmospheric, surface, and subsurface variables as input. In topographically complex watersheds, observations characterizing the environmental forcings required as input to these models (e.g., precipitation, temperature, wind speed, etc.) are sparse and often not representative. The complexity of the terrain, moreover, leads to gaps in or unavailability of radar-retrieved precipitation. As a result, in these watersheds data input to
hydrologic models is increasingly derived from the output of numerical weather prediction (NWP) models. The key advantage of these models is that they provide environmental forcings that are internally and physically consistent, and spatiotemporally continuous during the period of interest.

Challenges in the use of NWP models to derive hydrologic forcings include, however, the computational expense required to run these models and the discrepancies in spatial scales resolved by the atmospheric, versus land surface hydrology models
(Blöschl and Sivapalan, 1995; Wu and Li, 2009). Such discrepancies lead to significant uncertainties because the characteristic scales of processes and errors can be many orders of magnitude different between atmospheric and hydrologic models. Stated differently, a highly accurate weather prediction within a large region (e.g., the position of key synoptic phenomena is captured to within a few kilometers) can nevertheless lead to highly inaccurate hillslope-scape predictions when used to force a hydrologic model.



A fundamental challenge within the hydrologic sciences, therefore, is to characterize the relationship between the spatial resolution of numerical weather predictions used as input to hillslope-scale resolving models and the corresponding errors in predictions of quantities like SWE and soil moisture. Improving knowledge of the sensitivities of hydrologic predictions to the spatial resolution of weather prediction used as input to hydrologic models is of particular importance in the arid regions of the world that depend on snowmelt in upland mountain watersheds. In these regions, increasing precipitation variability and warming temperatures associated with climate change, coupled with increasing demands on water resources necessitate more accurate predictions of the spatiotemporal evolution of key hydrologic variables.

Even fewer of these studies examine the influence of the spatial resolution of atmospheric forcings on the multiple distributed and integrated variables simulated through integrated hydrologic models. Shao et al. (2001), for example, assessed the influence of the grid scales on both land surface and atmospheric processes. They suggest that surface-energy fluxes in areas where land surface heterogeneity is greatest exhibit the highest uncertainty. By contrast, Mölders and Raabe (1996) assessed the influence of aggregated land surface grid scale on atmospheric processes (e.g. evapotranspiration representing a key bottom boundary condition to atmospheric models) and suggest that the method used in aggregating land surface scale can reduce significantly the impact on atmospheric simulation results at coarse scale. Considering only the land surface, Maxwell (2007, as cited in Gentine et al., 2012) found that coarse scale representation of the surface hydrological processes in land surface modeling may have a significant effect on the area-averaged soil moisture estimates and introduce important biases because of scale. To our knowledge, only one study has performed a distributed spatial scale assessment that explicitly investigate the role of the forcing data resolution on snow modeling. Winstral et al. (2014) performed several modeling experiments in Dobson Creek catchment within Reynolds Creek Experimental Watershed by degrading 10 m forcing data, derived by geostatistically interpolating station data, progressively to a spatial resolution of 1500 m, while maintaining a snow model grid at 10 m resolution. The study suggests that significant effects on snow accumulation and melt occur due to the resolution of model forcing data, particularly when decreasing spatial forcing resolution. They postulate that topographic smoothness increases as the forcing resolution is degraded in complex terrain leads to increased snow accumulation in high-energy zones and less snow accumulation in low-energy zones. This correspondingly led to earlier melt and increased late-season water deficits. and also suggest that snow simulations performed with forcing scales of 250 m and larger, the errors increase. In summary, all these studies have identified multiple discrepancies on the estimations of hydrologic states and fluxes by applying combined scale modeling representations in either atmosphere and/or land surface and introduce systematic biases in integrated atmospheric and hydrologic modeling. Reconciling these discrepancies in scale while maintaining computational tractability, remains a fundamental challenge in surface hydrology.

In this study, we continue with the idea of the work introduced by Winstral et al. (2014) that through a series of carefully designed numerical modeling experiments, seek to advance fundamental understanding of the spatiotemporal expressions of coupled land-atmosphere processes in complex, snow-dominated watersheds. In our case, we introduce a framework that also incorporates the variability of the land surface and subsurface water storage which allow us to quantify the impact of the spatial resolution of the atmospheric forcings on simulated hillslope-scale hydrologic states and fluxes in a more integrated way.





The overarching goal of this study is to assess the impact of the atmospheric forcing scales on hydrologic state variables
such as snow water equivalent and soil moisture to potentially establish requirements for hydrologic modeling and observation
system design. The paper is organized as follow: (1) a description of the methods and an overview of the atmospheric and
hydrologic modeling framework, (2) a summary of the results and interpretation, (3) Discussion and conclusion. This study
attempts to emphasize the importance of the spatial and temporal meteorological scale to be used in any hydrology forecasting
and data assimilation in compensating for coarse resolution atmospheric forcing data, that also may potentially help to reduce
the computational and instrumentation cost, and at the same time attempt to improve the accuracy of surface and subsurface
hydrologic outputs.

## 2  Methods

### 2.1  Site

The study was conducted in Dry Creek Experimental Watershed (DCEW) which is used as a high-resolution testbed (Fig-
ure 1). DCEW is a hydrologic experimental site established in 1999. It is located 16 km northeast of the city of Boise, Idaho
with elevation ranging from 950 m to 2130 m. It is a semi-arid mountain front watershed, facing predominantly southwest.
Precipitation is dominated by snow in the upper basin and by rain in the lower elevations. Soil in Dry Creek is composed pri-
marily by coarse, sandy soil derived from in-situ weathering of granite (Gribb et al., 2009). Vegetation communities at lower
elevations are composed primarily of sagebrush, bitterbrush, mixed grasses and riparian vegetation, while at higher elevations
vegetation is dominated by coniferous evergreen trees, including Douglas Fir and Ponderosa Pine (Anderson et al., 2014).

### 2.2  Modeling Framework

The main components for the model execution within the area defined above are: (1) atmospheric processes represented by
weather research forecast model (WRF), a mesoscale numerical weather prediction model and (2) an open-source integrated
surface-subsurface hydrology model represented by ParFlow-CLM.

### 2.2.1  Atmospheric Model

The weather research forecast model (WRF) (Skamarock et al., 2005) with its advanced forecasting system version ARW
(advanced research WRF) is a numerical weather prediction (NWP) and atmospheric simulation system designed for research
and operational purposes. WRF is a community model developed and supported mainly by the National Center of Atmospheric
Research (NCAR). WRF-ARW integrates the compressible, nonhydrostatic Euler equations in order to deal with atmospheric
motion on a finer spatial scale. Those equations (which involve the equations of momentum and thermodynamic equation)
are written in flux form, in order to work either on the sphere by using geographical latitude-longitude coordinates or on a
conformal projection of the sphere by using Cartesian coordinates on a Mercator, Lambert or polar stereographic. An important
characteristic of WRF for our purposes is that it can be used in mesoscale regional atmospheric research as a Convection-



Permitting Model (CPM) (horizontal scale < 4km) that can explicitly resolve deep convection, which is a significant source of
precipitation, in a fine orographic scale (Prein et al., 2015). Within this context WRF simulations have been performed in several
studies as summarized in (Prein et al., 2015) for periods ranging from months to several years at horizontal scales ranging from
4 km to 0.8 km. In addition, WRF has been used with spatial domains that include complex terrain (e.g. mountain landscapes)
in a high-resolution scale such as Colorado Front Range at 1.3 km resolution (Mahoney et al., 2013), Idaho mountain ranges
spanning Owyhee mountains and Sawtooth mountains at 1 km resolution (Flores et al., 2016), and even at kilometer and sub-
kilometer scales (i.e., 333 m) on west-central Nevada (Horvath et al., 2012). All these studies have showed accurate results and
also allowed to obtain more detailed spatial representation of simulation outputs that serve as forcing for hydrologic modeling.

### 2.2.2 Hydrologic Model

For hydrologic modeling purposes, we selected the ParFlow-CLM model that is able to simulate both surface and subsur-
face hydrologic processes in an integrated way. ParFlow (Maxwell, 2013; Kollet and Maxwell, 2006; Jones and Woodward,
2001; Ashby and Falgout, 1996) is an open-source high spatial resolution surface-subsurface hydrology model which solves
simultaneously the three-dimensional Richards' equation and the kinematic wave equation for isothermal, transient and vari-
ably saturated flow using a Newton-Krylov method coupled to a multigrid preconditioned solver (Maxwell and Miller, 2005).
ParFlow has the advantage of an advance octree data structure which facilitates the watershed topography representation using
digital elevation models and geologic modeling of the subsurface. ParFlow also, is coupled with Common Land Model (CLM)
(Dai et al., 2003) where ParFlow computes the mass balance in the subsurface and CLM computes the mass and energy balance
at the land surface. Both models work simultaneously, by exchanging information of water fluxes between models at every time
step (Maxwell and Miller, 2005). More details of this coupled modeling system can be found in Maxwell and Miller (2005)
and (Kollet and Maxwell, 2008). ParFlow has been applied to several projects within the US as well as Europe and West Africa
ranging from local and regional watersheds applications (e.g., Gilbert et al., 2017; Fang et al., 2016; Maxwell et al., 2007) to
large scale continental (e.g., Condon and Maxwell, 2015; Keune et al., 2016), providing an excellent scalability of its solver
in distributed systems along with computationally accurate and efficient surface and subsurface flow solutions (Maxwell and
Miller, 2005).

### 2.3 Experimental setup

For this study we design and conduct a total of twelve numerical modeling experiments that seek to quantify the impact of
applied resolution of atmospheric forcings on simulated hillslope-scale hydrologic state variables that include two hydrologic
variables, Snow Water Equivalent and Soil Water Storage. The simulation experiments are arranged in such way to assess the
impact of four different atmospheric forcing resolutions (i.e., interpolated 30 m, 1 km, 3 km and 9 km) on three hydrologic
spatial resolutions commonly used for land surface modeling (i.e., 30 m, 90 m and 250 m). Table 1 illustrates the set of
simulation experiments performed in this study. In order to setup all the experiments two major phases were needed: (1) the
atmospheric forcing generation (input data to Parflow) and hydrologic model parameter adjustment and (2) Scale assessing
framework development and implementation for the atmospheric and land surface multiscale modeling.





### 2.3.1   Atmospheric forcing generation and hydrologic model parameter adjustment

Surface hydrometeorological data required as input to the hydrologic model are obtained from the WRF model. The model produces distributed meteorological forcing data at several scales to feed the land surface-subsurface component represented

by ParFlow and CLM, incorporating consistent environmental forcing data distributed over the landscape that consider remote and ungauged areas. After synthesizing the hydrometeorological forcings to ParFlow using WRF, we adjusted the Manning roughness value on the study site, using the finest land surface spatial scale, in order that the model outputs match acceptably the observed data. This process was manually performed and the outputs targeted were soil moisture and streamflow, and they were compared with observational soil moisture and streamflow data obtained by sensors and gauges distributed within Dry

Creek Experimental Watershed. The value suggested was 0.000094 $h/m^{1/3}$. Initial conditions for Parflow were obtained by performing a set of drainage experiments that allow us to find an initial moisture state for a particular time that matches a real condition in the dry season.

### 2.3.2   Scale assessing framework development and implementation for the atmospheric and land surface multiscale modeling

A scale assessment framework that compares the performance of model simulations against a model-derived "true" state was developed by using the previous atmospheric input generation and hydrologic model parameter adjustment. Figure 2 describes the overarching process of scale assessing framework adapted to our case. This framework consists on the generation of a reference hydrologic state focusing primarily on snow water equivalent and soil water storage for a certain period. In data assimilation and inverse theory studies this methodology is often referred to as a Nature Run or Truth and for this study the

experimental simulation performed with highest spatial resolution (i.e., interpolated 1 km to 30 m atmospheric forcing scale and 30 m hydrologic scale) is considered as nature run. Then, several experiments were run, maintaining constant spatial resolution of the land-surface and using multiple meteorological forcing resolutions decided arbitrarily (see Table 1). These types of runs are often called perturbation runs which then are compared to the nature run. The impact of the scale resolution was measured by estimating the mean error (ME) and the root mean square error (RMSE) between the nature (observations)

and perturbation (simulated) runs. The bias, or mean error is computed as,

$$ME = \frac{1}{N}\sum_{i=1}^{N}(\delta_{E,i,j} - \delta_{T,i,j}) \tag{1}$$

and the root mean squared error is computed as,

$$RMSE = \sqrt{\frac{1}{N}\sum_{i=1}^{N}(\delta_{E,i,j} - \delta_{T,i,j})^2} \tag{2}$$





where $\delta_{E,i,j}$ is the hydrologic variable to assess for each experiment at the coordinates $(i,j)$ in the modeling grid and $\delta_{T,i,j}$
represent the hydrologic variable to assess for the control run, $N$ represents the number of samples in time (time series length)
assessed.

## 2.4   Hydrologic variables assessed

In this study, we assessed two hydrologic variables of importance that represent surface and subsurface processes: total soil
water storage and snow water equivalent. These processes and corresponding mass and energy fluxes within model grid cells
and external forcings are depicted in Figure 3. The total soil water storage in a soil column is represented by the sum of the
product of saturation, porosity and the layer thickness as,

$$\Theta = \sum_{k=1}^{L} S_{i,j,k}\phi_{i,j,k}\Delta z \tag{3}$$

where $\Theta$ represents soil water storage, $S$ is the water saturation in layer located at coordinates $(i,j,k)$, $\phi$ is the porosity in
layer $k$ and $\Delta z$ corresponds to the layer thickness. The water saturation in ParFlow model is estimated from the pressure field
(equation 3.2), using the mixed form of the Richards equation (Maxwell and Miller, 2005) as,

$$\frac{\partial(s(p))\rho\theta)}{\partial t} - \nabla\left[\frac{k(x)k_r(p)\rho}{\mu}(\nabla p - \rho g \nabla z)\right] = q \tag{4}$$

the variable $s(p)$ is the water saturation at a hydraulic pressure $p$ while the variables $\theta$, $k(x)$ and $k_r(p)$ are the effective
porosity, the absolute permeability and the relative permeability of the medium, $\rho$ and $\mu$ represent the density and dynamic
viscosity of the water. Snow water equivalent, represents the amount of water that results if snowpack is melted by unit of area
(DeWalle and Rango, 2008; Armstrong and Brun, 2008). It can be simply expressed as the measurement of snow depth $h_s$ and
the density ratio between snow $\rho_s$ and liquid water $\rho_w$ as,

$$SWE = h_s \frac{\rho_s}{\rho_w} \tag{5}$$

However, in CLM model the amount of water in a snowpack is determined by the conservation of mass and energy of the
snowpack within a control volume, neglecting horizontal fluxes but considering vertical neighbors (i.e., snow layers). The mass
balance is represented in terms of mass of water $w_{liq}$ and mass of ice $w_{ice}$ which the temporal variation of mass of water in
such control volume is driven by liquid phase, ice phase water fluxes and changes in water phase $M_{il}$ due to melt (melting rate)
as follows,

$$\frac{\partial w_{liq,k}}{\partial t} = (q_{liq,k-1} - q_{liq,k}) + (M_{il}\Delta z)_k \tag{6}$$





$$\frac{\partial w_{ice,k}}{\partial t} = (q_{ice,k-1} - q_{ice,k}) - (M_{il}\Delta z)_k \tag{7}$$

The energy balance in snowpack is represented by the conservation of energy equation which is defined as the change in stored energy within a control volume (snow layer) equal to the net energy flux across the volume surface as follows,

$$\sum_{m=i,l} [\rho z_m c_m \theta_m]_k \Delta z_k \frac{\partial T_k}{\partial t} = R_{n,k} - [L_f M_{il}\Delta z]_k - H - L_v E + \left[\lambda \frac{\partial T}{\partial t}\right]_{z_{h,k-1}}^{z_{h,k}} \tag{8}$$

where $t$ is the time, $T_k$ is the average temperature of the layer $k$, $\rho_m$ , $c_m$ and $\theta_m$ are the density, specific heat and the partial volume of the water $l$ and ice $i$. $L_f$ and $L_v$ latent heat of fusion for ice and latent heat of evaporation, $\lambda$ is the is the
thermal conductivity of snow, and $R_{n,k}$, $H$ and $L_v E$ are radiative, latent and sensible heat fluxes. The snow component in CLM is represented by up to five snow layers and the momentum, latent heat and sensitive heat fluxes between the atmosphere at reference height (i.e., 10 m) and snow surface, are derived from Monin-Obukhov similarity theory (Monin and Obukhov, 1954).

### 2.5  Data requirement and model configuration

The ParFlow-CLM model requires hourly forcing inputs of precipitation, temperature, pressure, wind speed, incoming short and longwave radiation, and specific humidity. The forcing input data are generated by WRF models and retrieved as a data-subset from the 30-Year, Multi-Domain High-Resolution Climate Simulation Dataset for the Interior Pacific Northwest and Southern Idaho project (Flores et al., 2016). Data used to verify model outputs include observations of hourly and daily average discharge, soil moisture, and snow water equivalent are located at multiple sites in DCEW. Such observation data were
retrieved from DCEW in water year 2009 at https://earth.boisestate.edu/drycreek/data/.

To develop the computational grid for the ParFlow-CLM model the following data were needed: (1) digital elevation data defining watershed topography, (2) the spatial distribution of soil types (e.g., surface texture), and (3) the spatial distribution of land cover data. The digital elevation data for the domain area was retrieved from the National Elevation Dataset (Gesch et al., 2002) data source at 1/9 arc-second spatial resolution (30 m), while other domain spatial scales (i.e., 90 m and 250 m)
were retrieved by upscaling such data sources using QGIS raster alignments tools (QGIS Development Team et al., 2014) with a bilinear resampling method. Soil types were retrieved from Soil Survey Geographic Database (SSURGO)(Soil Survey Staff, 2014) and the soil texture classifications were determined based on the percent of sand and clay by using NRCA's soil texture calculator (Natural Resources Conservation Service, 2014). Hydraulic and soil moisture characteristic parameters (i.e., Van Genuchten parameters) were retrieved from Leij (1996) and Simmers (2005) while land cover datasets, were retrieved from
the National Land Cover Database (NLDAS) (Homer et al., 2015). Since CLM model requires the International Geosphere-Biosphere Programme (IGBP) land cover classification, primarily retrieved from 500 m spatial resolution MODIS data, each NLCD classification is approximated to the corresponding IGBP classification required by CLM model. This approach takes





advantage of using a finer resolution of NLDAS land cover dataset and the similitude existing between MODIS and NLDAS datasets.

The computational soil grid in ParFlow uses terrain following grid formulation (Maxwell, 2013), which has the advantage of solving all the governing equations in a near ground complex terrain and fine spatial discretization, more efficiently. The soil profile and layer thicknesses are selected according to the ParFlow-CLM modeling requirements. Figure 4 illustrates the soil layers configuration as well as the thickness value for each layer and total soil depth for the modeling domain. The top soil layer was set to 1 m depth for the entire domain followed by 19 m of bedrock. The first two top soil layers in ParFlow are set

with small values (i.e., 0.05 m each) of thickness to allow efficient flux exchange with CLM model. The subsequent soil layers are distributed by equal valued thicknesses (i.e., 0.1 m) up to reach 0.8 m depth. The transition between soil and bedrock are set as 0.2 m from the soil part and 1 m from the bedrock part followed by 9 layers of 2 m each up to reach 20 m depth. The set of hydraulic and soil moisture characteristic parameters selected for each soil texture are shown in Table 2 and are used for all the simulation experiments.

### 250   2.5.1   Forcing data and initial condition generation

The atmospheric forcing data generated by the WRF model, are spatially distributed (9 km, 3 km, 1 km and 30 m interpolated from 1 km) and available at hourly resolutions during water year 2009. Since we performed different experiments at 30 m, 90 m and 250 m land surface spatial resolution, the forcing data are rescaled to the corresponding land surface model grid using a nearest neighbor interpolation algorithm to maintain the atmospheric spatial scale of the original data output of WRF. Another

set of three experiments are rescaled from 1km WRF grid to the 30 m ParFlow grid using bilinear interpolation in order to provide a set of meteorological forcings that are smooth in space at the scale of the ParFlow model.

Drainage experiments are performed for all land surface scales in order to find a reasonable initial condition of water pressure to initialize the model (Figure 5). The drainage experiment consisted of dampening the domain with a small and constant value of recharge in order to reach groundwater equilibrium (Figure 5-A). At this stage, the lateral flows on each cell are turned off

to avoid formation of streams and ponds. Afterwards, we allowed the watershed to drain by turning on the lateral flows. This drainage experiment allows us to estimate continuous pressure head and saturation fields for all the cells in the domain and also allows groundwater to converge and discharge through the valleys and form streams (Figure 5-B). During drainage, we allow groundwater storage and streamflow to decrease until the streamflow matches a reference value of observed streamflow at the watershed outlet in the dry season. The value to match was $0.0031 m^3/s$ which corresponds to the streamflow value obtained

at the end of September 2008 (Figure 5-C). The states of the ParFlow model, pressure head and saturation, at this discharge during the drainage experiment are then retained. We use these retained states as the initial conditions for the complete set of experiments depicted in Figure 6. The top layer of head pressure for land surface domains at 30 m, 90 m and 250 m spatial scale are shown in Figure 6 with a sample of distributed atmospheric forcing at scale of 30 m, 1 km, 3 km and 9 km.



### 2.5.2 Baseline run

A baseline run (Nature Run) is obtained by performing a short-term simulation, comparing simulated streamflow, soil moisture, and snow water equivalent to observed values in DCEW, and a set of parameters principally in the soil component of the model, were selected from Leij (1996) to try to match the simulated and observed values. The nature run coincided with the experiment with the finest hydrologic resolution and the finest interpolated atmospheric forcing scale (i.e., 30 m hydrologic resolution and 30 m bilinear interpolated forcing resolution).

Once Parflow and CLM finish their simulations, the model performance is assessed where data variables are available for verification. Figure 7 shows for example the model performance of soil moisture with respect to observed data. Simulated values of soil moisture are either underestimated and/or overestimated in some periods during 2009 water year. These discrepencies arise primarily due to effects of rainfall events simulated by WRF that do not coincide in time with the observations, as well as uncalibrated hydraulics parameters in the soil component. However, calibration to improve model predictability in the baseline

run is not the main purpose of this study, but rather to use it as a reference to compare with the other experiments and determine the impact of applied resolution of atmospheric forcings on simulated hillslope-scale hydrologic state variables.

## 3 Results

Once all of the numerical modeling experiments are performed according to the modeling assessment framework developed, all the experiments were compared to the baseline run in order to verify spatial error patterns due to variations in spatial scale.

The spatially distributed results for all the experiments are presented in terms of soil water storage integrated vertically in the soil column (i.e., 1 m soil column), and SWE, which allows us to assess the subsurface and surface water balance sensitivity to changes in atmospheric forcing scales.

### 3.1 Spatially distributed RMSE for SWE and soil water storage

Analysis of Figure 8 shows the effects of variation in the spatial resolution of atmospheric forcing inputs on SWE across

a corresponding range of ParFlow spatial resolutions in DCEW. For all land surface resolutions, the RMSE increases with decreasing resolution in the atmospheric forcing. In terms of magnitude of the RMSE, correspondingly, increasing land surface scale and atmospheric scale increases the magnitude of errors in SWE. Local effects of forcing data are particularly pronounced in simulations at 30 m land surface spatial resolution in which high magnitudes of RMSE are located in borders and/or corners defined by the corresponding atmospheric grid resolution. These patterns are due to the comparison of the effects of coarse

atmospheric forcing resolution and finest ones to simulated SWE. High spatial resolution modeling (i.e., nature run) produce of high gradients of SWE across in the modeling domain and when it is compared to effects caused for aggregated forcing (i.e., 1 km, 3 m and 9 km) on SWE, these types of artifacts are generated concentrating less error around the middle of the coarse grid cell and larger along the boundaries.





On the other hand, the errors in the soil water storage shown in Figure 9, shows slightly different RMSE patterns at 30
m land surface model resolution in contrast to SWE cases. In these cases, the atmospheric patterns caused particularly in
SWE as shown in Figure 8, are now combined to topographic effects on streamflow in which higher moisture converge to
lower elevated areas. On the other hand, comparisons of 90 m and 250 m land surface spatial resolution to the nature run show
significant topography and aspect patterns on the spatial pattern in RMSE of soil water storage. Specifically, there are significant
topographic effects on soil water storage patterns and large effects on flat areas, such as riparian areas. More general statistics
that show RMSE variability expressed above between experiments and the nature run are also illustrated in tables 2 and 3 in
which minimum, maximum, mean and standard deviation of RMSE measurements for SWE and soil water storage variables
are shown respectively. These global statistics, particularly the mean of the spatial RMSE, indicate the drastic incremental
changes in magnitude, specially for SWE, when degrading the hydrologic scale (i.e., from H1 to H2) as well as indicate the
incremental change when degrading the atmospheric scale while maintaining the hydrologic scale.

**3.2 Spatially distributed bias for SWE and soil water storage**

Analysis of bias for SWE as shown in Figure 10 reveals (similar to RMSE analysis) that the most positive biases are found
around the borders of the associated forcing grid at 30 m land surface resolution. Also, decreasing in both atmospheric and land
surface spatial resolutions, it is revealed that topography and slope aspect have significant impact on simulated SWE. Visual
inspection on simulations over 90 m land surface resolution shows that the most positive biases are associated with north
aspects and lower elevation areas (case of land surface simulated with 9 km atmospheric resolution). Conversely, negative
biases are mostly associated with south facing aspects and higher elevations.

Visual inspection of biases for soil water storage ( 11) reveals that most positive biases are associated with the borders of
the associated forcing grid in the 30 m land surface resolution simulation. However, for atmospheric resolutions over 3 km,
both positive and negative biases are associated with valley bottoms. On the other hand, decreasing both atmospheric and land
surface spatial resolution, shows that topography has significant impact on the spatial distribution of bias in soil water storage.
On simulations performed with 90 m land surface resolution and above, it is found that the mean of negative and positive biases
are associated with stream and riparian areas respectively. More general statistics that show ME variability expressed above
between experiments and nature run are also illustrated in tables 5 and 6 in which minimum, maximum, mean and standard
deviation of ME measurements for SWE and soil water storage variables are shown respectively. Similarly as RMSE analysis
in the previous subsection, drastic changes can be visible in the mean of the ME, specially when degrading the hydrologic scale
(i.e., from H1 to H2).

**4 Discussion**

This study is meant to develop a spatial scale assessment framework that can be used to inform the discrepancies on hydrologic
modeling outputs due to the use of coarse scale atmospheric forcing data. By performing a set of synthetic experiments, this
study has revealed the importance of choosing a specific spatial scale in both atmospheric and land surface to maintain con-





sistency of the modeling outputs in comparison to observations as well as computational tractability. We found that modeling at high resolution land surface scales, the processes assessed are locally driven by atmospheric forcings downscaled to the corresponding land surface scale. However, as decreasing in land surface resolution (i.e., coarser spatial resolution) the effects of the topography are more influential on the variables assessed.

As seen in figures 8 and 10, simulations for snow variables (i.e., SWE) at high resolution land surface scale of (i.e., 30 m.), the variability of bias and RMSEs are particularly controlled by the atmospheric grid, revealing more local effects over the spatially distributed variable modeled. This leads to large errors in SWE, concentrated on opposite sides or corners of the atmospheric grid cells produced by gradients of magnitude of SWE existent on the base resolution of the nature run or reference, where atmospheric scale is interpolated from 1 km to 30 m. The scale artifacts as depicted might demonstrate that

snow processes are not only dependent on orographic characteristics at that specific spatial scale, but also are highly dependent on the complex interaction between mass and energy inputs coming from the atmospheric processes and local topography (Grünewald et al., 2014). Bias and RMSEs in simulations at coarse land surface scales (i.e., 90 m. and 250 m.) are controlled more by the topography than the atmospheric grid and clearly reveal the effects of the topography on snow processes at these scales in which larger errors in SWE estimation in the entire time series simulation, occur in upper areas of the watershed where

naturally snow precipitation and accumulation are considerable. Looking at slope aspects it is observed that most north-facing slope areas show less error compared to south-facing slope areas. This can be attributable to the fact that thermal forcings are more significant in south-facing slopes, particularly in the snow ablation process (Pomeroy et al., 2003) which is also becoming even more sensitive to the climate warming (López-Moreno et al., 2014).

In simulations for soil water storage at high land surface model resolutions (i.e., 30 m), the RMSEs are primarily controlled

by atmospheric grid in a similar manner to SWE. Since snow is one of the main sources of soil moisture in mountain environments (Marks et al., 1999), spatial patterns in soil water storage become highly dependent of inputs of water controlled by snow melting processes (Williams et al., 2009). These are, in turn, controlled by atmospheric forcing effects and therefore, such effects can be propagated directly to the soil moisture. On the other hand, at land surface model resolutions of 90 m and 250 m, larger errors in soil water storage occur in riparian areas of the watershed where accumulation of moisture due to topography

and vegetation are considerable. Particularly on north facing slopes there is less error compared with south facing slopes due to large radiation effects on snowmelt dynamics, which are also propagated to the water inputs to the soil. Such coarse scale atmospheric effects on soil water storage reinforce some previous studies focused in hydrologic responses performed by Wood et al. (1988) in which a watershed domain is represented in a catchment scale or Representative Elementary Areas (REA). The authors found that REA are strongly influenced by topography in which increases in the variability of rainfall input and soil

between sub-catchments, causing significant changes in surface runoff between sub-catchments.

On the other hand, Molnar and Julien (2000) state that finer land surface spatial scales are important for short duration events and coarser spatial land surface spatial scales are suitable for long precipitation events. A similar study performed by Bormann (2006) suggests that aggregating input (atmosphere) as well as decreasing spatial scale at land surface (increasing grid size) may cause effects in streamflow simulation results. Additionally, coarse land surface spatial scales must preserve

important information of land use classes since they are determinant in vertical processes such as evapotranspiration. His study



suggests that the statistics of the water balance are not significantly dependent on the land surface spatial resolution. However, this requires high quality input data and posteriorly optimized to the best resolution. Finally, Vivoni et al. (2005) suggest that domain resolution is a critical step for a modeling application. Decreases in terrain resolution may lead to potential increments of uncertainty in the hydrologic response. In addition, their study suggests that saturation dynamics are better captured at high

resolutions, particularly in the vicinity of streams (riparian areas) since these areas contribute directly and significantly to runoff and evapotranspiration processes.

Limitations of this study lie in the type of atmospheric and hydrologic modeling platform selected as well as the climatological and topographic characteristic study area. In addition, the selection of highly detailed physically based atmospheric and hydrologic models is of great importance in this type of assessment. The selection of physical parameterizations in both the

atmospheric and hydrologic models must give enough confidence that coupled complex land surface and subsurface processes are well represented and consistent to changes in spatial scale. This study opens avenues for future research that investigate the effects of atmospheric forcing scales by using several other coupled atmospheric-land surface and subsurface models. Potential pairs of models of interest include the coupled version of WRF-hydro, ParFlow-ARPS, ParFlow-Cosmo, and others. This study also suggests potential value in investigating scale effects in modeling systems applied to different types of climate

and/or topography

## 5   Conclusions

In this study, we introduce a multi scale modeling assessment framework which was useful to investigate and quantify the impact of the spatial resolution of the atmospheric forcings (i.e., wind speed, air temperature, relative humidity, incoming longwave and shortwave radiation) on simulated hillslope-scale hydrologic states and fluxes such as snow water equivalent and

soil water storage of coupled land-atmosphere processes in complex, snow-dominated watersheds.

The spatial distributions of errors (i.e., MEs and RMSEs) found in simulations for SWE at high resolution land surface scale (i.e., 30 m) are particularly driven by the atmospheric grid, revealing local effects over the spatially distributed variable modeled. Simulations performed with coarse land surface scales (i.e., 90 m and 250 m) as well as with coarse atmospheric spatial scales, has errors that are driven more by topography than the atmospheric grid and clearly reveal the effects of topography and

slope aspect on snow processes. However, both play an important role on the effect of the spatial SWE magnitude across the watershed.

The spatially distributed errors found in simulations for soil water storage at the high resolution land surface scale are driven by the atmospheric grid scale, similar to corresponding patterns in SWE errors. However, simulations performed for land surface scales at 90 m and 250 m as well as with coarse atmospheric spatial scales, larger errors in the entire time series

simulations occur in riparian areas of the watershed, where accumulation of moisture due to flat areas in the topography is significant.

In this study, we have emphasized the importance of the effects caused by atmospheric forcing scales on simulated hydrologic state variables such as snow water equivalent and soil moisture in the regions that depend on snowmelt. The study has also



improved the knowledge of the sensitivities of hydrologic predictions to the spatial resolution of weather prediction data used

as input to hydrologic models and helped to characterize the relationship between the spatial resolution of numerical weather

predictions used as input to hillslope-scale resolving models.

*Code availability.*   Codes can be found in the public GitHub repository at: https://github.com/miguelaguayo/ParFlow-CLM-Scripts/

# 6   Figures and Tables

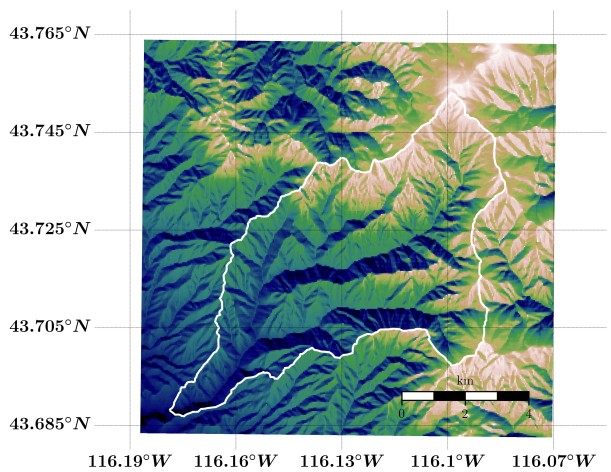

**Figure 1.** Dry Creek Experimental Watershed study area.





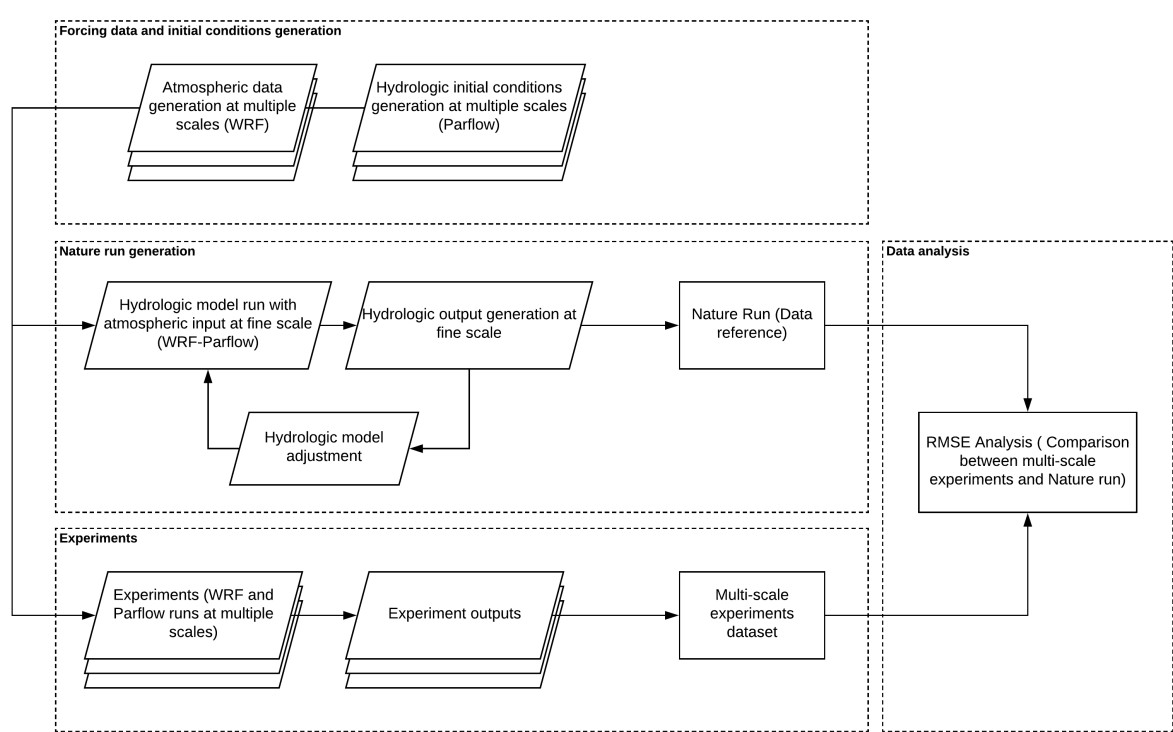

**Figure 2.** Multiscale modeling assessment framework.





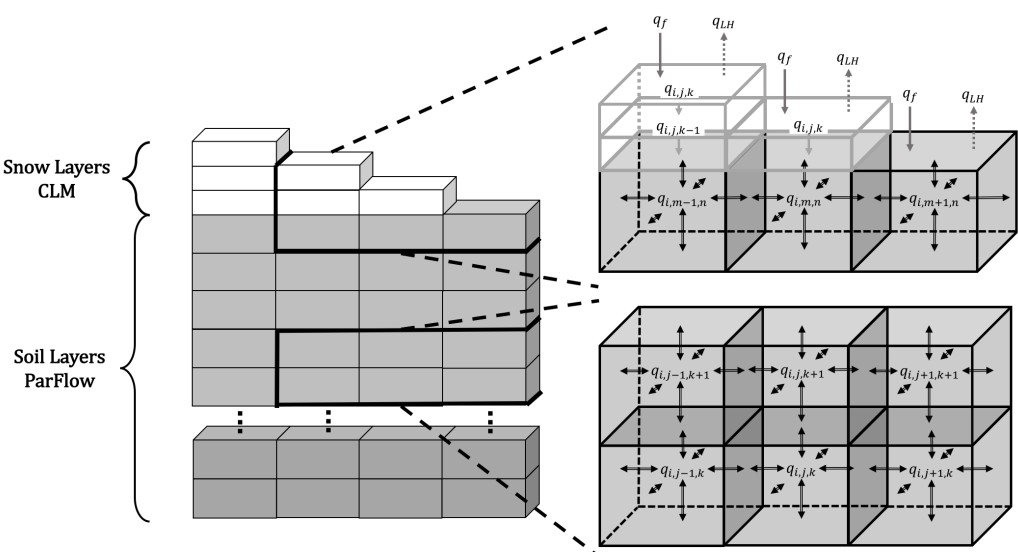

**Figure 3.** Schematic cell representation of surface and subsurface layers in ParFlow-CLM. Variables $q_f$ represent downward energy fluxes from forcings and $q_{LH}$ upward energy fluxes (sensible and latent) respectively, from snow and soil

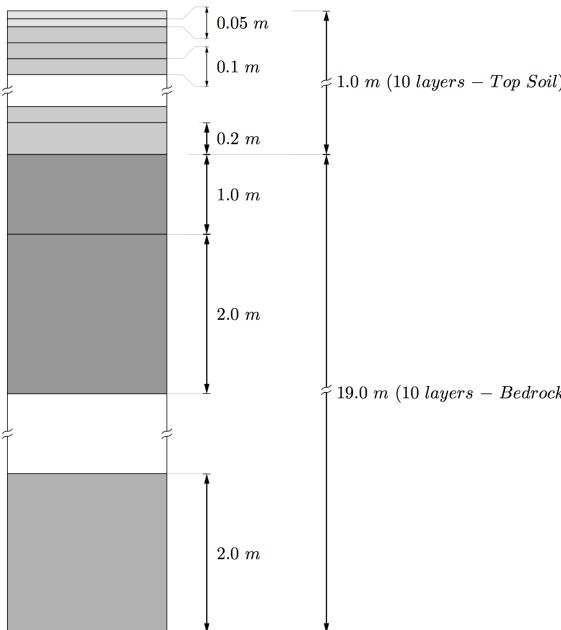

**Figure 4.** Detailed soil layers configuration for the modeling domain used in ParFlow-CLM simulations.





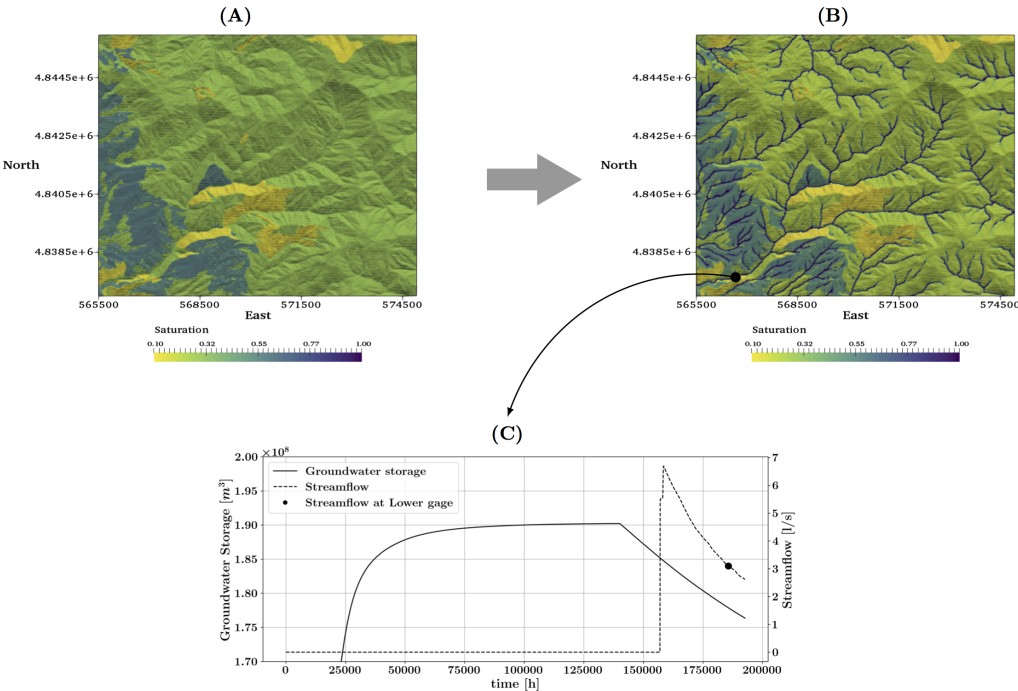

**Figure 5.** Soil moisture initial conditions generation process.

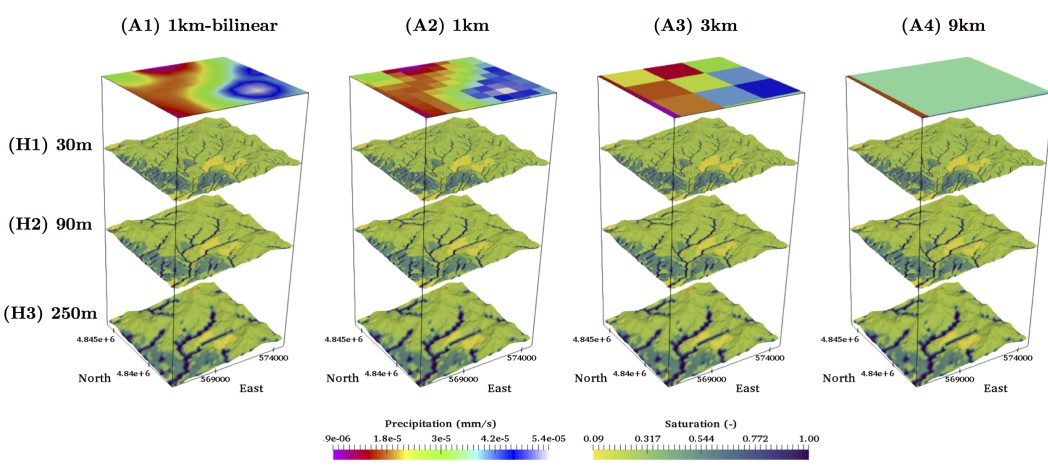

**Figure 6.** Soil moisture initial conditions and an example of distributed atmospheric data generated by WRF model.





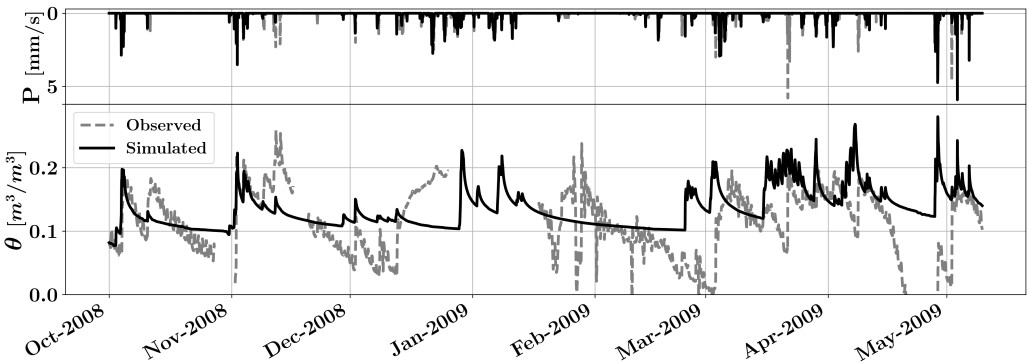

**Figure 7.** Soil moisture at 2 cm observed simulated time series at lower south station in DCEW. Precipitation data is retrieved from Lower Weather station (see Figure 1).

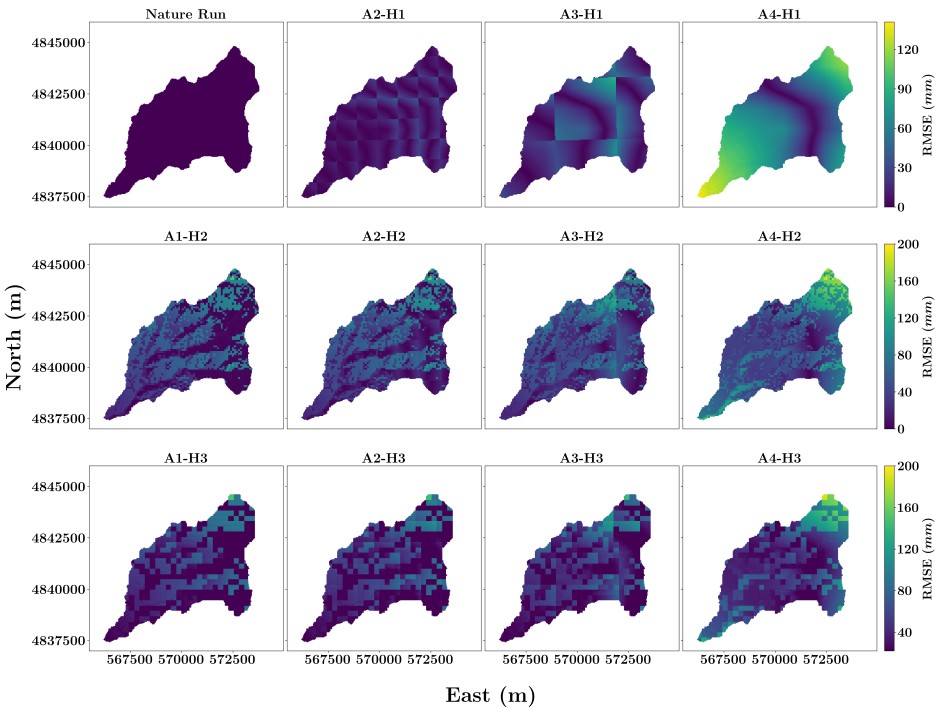

**Figure 8.** Spatially distributed RMSEs for SWE in water year 2014.





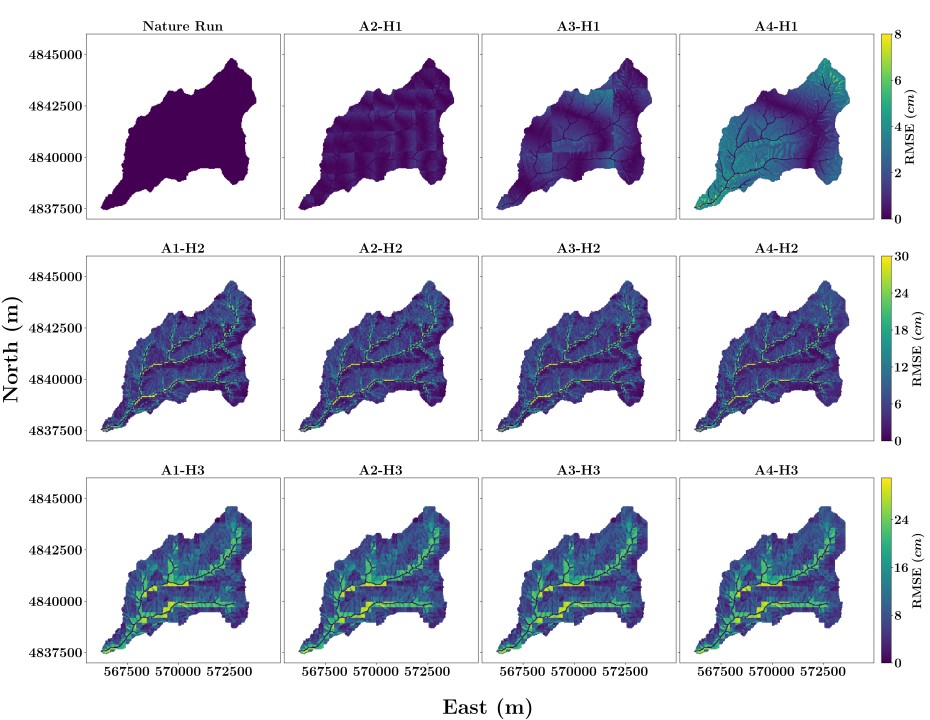

**Figure 9.** Spatially distributed RMSEs for soil water storage in water year 2014.

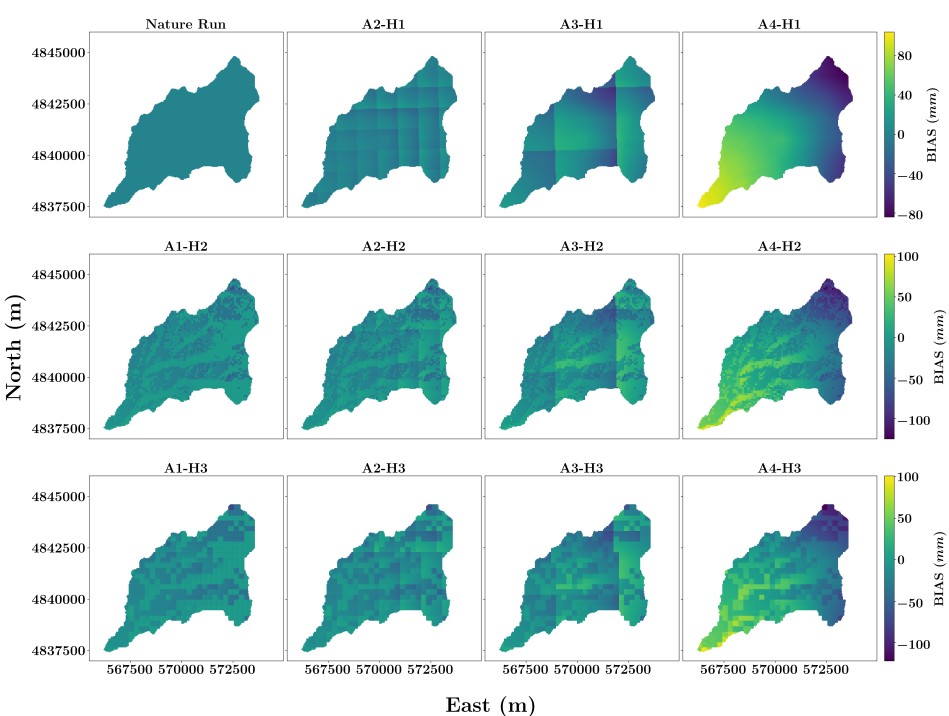

**Figure 10.** Spatially distributed bias for SWE in water year 2014.

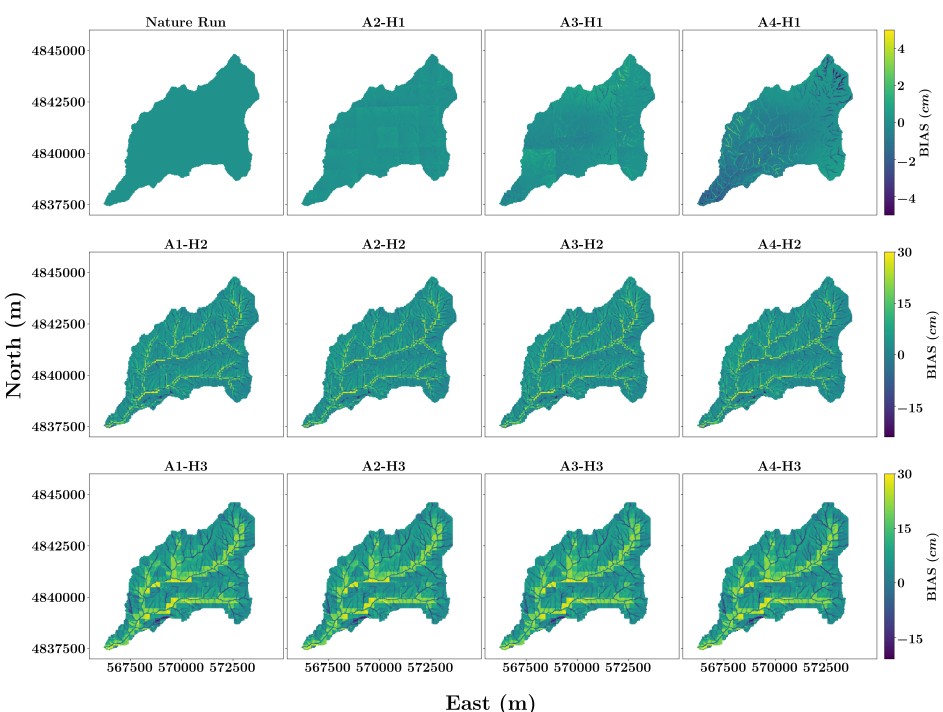

**Figure 11.** Spatially distributed ME for soil water storage in water year 2014.





**Table 1.** Simulation experiments and nomenclature.

| Hydrologic Resolution | Atmospheric Forcing Resolution | | | |
|---|---|---|---|---|
| | A1 (H m)[a] | A2 (1 km) | A3 (3 km) | A4 (9 km) |
| H1 (30 m) | A1-H1[b] | A2-H1 | A3-H1 | A4-H1 |
| H2 (90 m) | A1-H2 | A2-H2 | A3-H2 | A4-H2 |
| H3 (250 m) | A1-H3 | A2-H3 | A3-H3 | A4-H3 |

[a] Bilinearly interpolated to hydrologic scale from 1 km.

[b] Simulation reference (Nature run).

**Table 2.** Summary of soil parameters selected from Leij (1996) and Simmers (2005) and applied to the DCEW study domain.

| Soil Texture | Parameters | | | | |
|---|---|---|---|---|---|
| | $\theta_r(m^3/m^3)$ [a] | $\theta_s(m^3/m^3)$ [b] | $\alpha(m^{-1})$ [c] | $n(-)$ [d] | $K_s(m/hr)$ [e] |
| Loamy sand | 0.057 | 0.41 | 12.4 | 2.28 | 0.1459 |
| Sandy loam | 0.065 | 0.41 | 7.5 | 1.89 | 0.0442 |
| Sandy clay loam | 0.100 | 0.39 | 5.9 | 1.48 | 0.0131 |
| Loam | 0.078 | 0.43 | 3.6 | 1.56 | 0.0104 |
| Clay | 0.068 | 0.38 | 0.8 | 1.09 | 0.0020 |

[a] Saturation residual.

[b] Saturated water content.

[c] Parameter inversely related to the air entry value.

[d] Parameter that determine the water retention curve shape.

[e] Saturated hydraulic conductivity.

**Table 3.** Summary of RMSEs for SWE (mm).

| Experiment | Metrics | | | |
|---|---|---|---|---|
| | Min | Max | Mean | SD |
| A2-H1 | 0.50 | 44.55 | 9.04 | 6.84 |
| A3-H1 | 0.36 | 91.38 | 22.97 | 16.33 |
| A4-H1 | 1.75 | 140.69 | 60.64 | 33.14 |
| A1-H2 | 0.43 | 153.92 | 33.60 | 30.18 |
| A2-H2 | 0.53 | 161.90 | 36.48 | 28.12 |
| A3-H2 | 0.40 | 147.63 | 42.03 | 25.26 |
| A4-H2 | 1.64 | 196.09 | 57.26 | 33.12 |
| A1-H3 | 0.51 | 152.97 | 34.40 | 29.82 |
| A2-H3 | 0.55 | 149.28 | 37.30 | 27.88 |
| A3-H3 | 0.83 | 144.61 | 41.61 | 25.38 |
| A4-H3 | 1.28 | 192.16 | 56.53 | 32.99 |





**Table 4.** Summary of RMSEs for soil water storage (cm).

| Experiment | Metrics | | | |
|---|---|---|---|---|
| | Min | Max | Mean | SD |
| A2-H1 | 0.00 | 1.99 | 0.41 | 0.31 |
| A3-H1 | 0.00 | 5.64 | 0.95 | 0.69 |
| A4-H1 | 0.00 | 10.71 | 2.13 | 1.26 |
| A1-H2 | 0.00 | 29.46 | 6.08 | 5.01 |
| A2-H2 | 0.00 | 29.46 | 6.10 | 5.00 |
| A3-H2 | 0.00 | 29.46 | 6.15 | 4.96 |
| A4-H2 | 0.00 | 29.46 | 6.22 | 4.89 |
| A1-H3 | 0.00 | 30.08 | 9.50 | 6.42 |
| A2-H3 | 0.00 | 30.08 | 9.53 | 6.40 |
| A3-H3 | 0.00 | 30.08 | 9.56 | 6.39 |
| A4-H3 | 0.00 | 30.09 | 9.54 | 6.37 |

**Table 5.** Summary of ME for SWE (mm).

| Experiment | Metrics | | | |
|---|---|---|---|---|
| | Min | Max | Mean | SD |
| A2-H1 | -28.84 | 24.08 | -1.11 | 7.38 |
| A3-H1 | -62.19 | 43.10 | 1.38 | 19.10 |
| A4-H1 | -83.82 | 104.52 | 12.85 | 46.35 |
| A1-H2 | -75.61 | 10.19 | -14.00 | 13.90 |
| A2-H2 | -85.53 | 21.56 | -14.12 | 15.20 |
| A3-H2 | -77.82 | 43.84 | -11.89 | 21.43 |
| A4-H2 | -125.01 | 103.13 | -8.75 | 39.69 |
| A1-H3 | -75.70 | 12.73 | -14.24 | 13.98 |
| A2-H3 | -72.39 | 23.73 | -14.94 | 15.00 |
| A3-H3 | -75.60 | 38.29 | -12.47 | 20.65 |
| A4-H3 | -121.53 | 101.34 | -8.47 | 39.22 |



**Table 6.** Summary of ME for soil water storage (cm).

| Experiment | Metrics | | | |
| --- | --- | --- | --- | --- |
| | Min | Max | Mean | SD |
| A2-H1 | -0.91 | 1.35 | 0.07 | 0.16 |
| A3-H1 | -2.84 | 3.64 | 0.06 | 0.41 |
| A4-H1 | -5.89 | 4.95 | -0.39 | 0.77 |
| A1-H2 | -22.85 | 29.12 | 3.95 | 5.77 |
| A2-H2 | -22.99 | 29.12 | 3.96 | 5.77 |
| A3-H2 | -23.09 | 29.12 | 3.99 | 5.77 |
| A4-H2 | -24.31 | 29.12 | 3.89 | 5.82 |
| A1-H3 | -22.80 | 29.88 | 7.51 | 7.71 |
| A2-H3 | -22.93 | 29.88 | 7.53 | 7.70 |
| A3-H3 | -22.74 | 29.88 | 7.57 | 7.69 |
| A4-H3 | -21.84 | 29.88 | 7.45 | 7.76 |



*Competing interests.* The authors declare they have no conflict of interest.

*Acknowledgements.* This work was supported by grants from National Science Foundation award IIA-1329513 Collaborative Research: Western Consortium for Watershed Analysis, Visualization, and Exploration (WC-WAVE); NASA award NNX15AD56G, "Multiple frequency active microwave remote sensing for snow water equivalent retrieval from space: A data assimilation approach." and NASA award NNX14AN39A, "Monitoring Earth's Hydrosphere: Integrating Remote Sensing, Modeling, and Verification". We would like to thank Katelyn Fitzgerald from NCAR and Dr. Matt Masarik for their support on WRF modeling and simulations. This research made use of the resources
of the High Performance Computing Center at Idaho National Laboratory, which is supported by the Office of Nuclear Energy of the U.S. Department of Energy and the Nuclear Science User Facilities under Contract No. DE-AC07-05ID14517. We also thank Ben Nickell and Eric Whiting from Idaho National Laboratory for their support on high performance computing.



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
