# Peer review of "Examining cross-scale influences of forcing resolutions in a hillslope-resolving, integrated hydrologic model"

_Hydrology and Earth System Sciences, 2020_

## Referee Comment (RC1) · Anonymous Referee #1 · 18 Nov 2020

Review of **Examining cross-scale influences of forcing resolutions in a hillslope-resolving, integrated hydrologic model** by Aguayo et al.

The study of Aguayo et al. focusses on assessing the influence of different spatial scales of atmospheric forcing and hydrologic modeling. The Weather Research Forecasting (WRF) model is used and interpolated to different spatial scales to force the ParFlow hydrological model, which is also run on different spatial scales. The authors conclude that the spatial distribution of errors in SWE are at high land surface scale driven by the atmospheric resolution, whereas with coarser land surface scales and atmospheric scales these are more driven by the topography. The total soil water storage showed similar errors, but at coarser scales larger errors due to the topography were found in for example riparian areas.

Generally, the study is concise and has a clear goal, with a well-structured lay-out. However, the list of minor issues is still rather long, that mainly relate to sentences that were unclear to me and may need rephrasing. I also have several more major issues, and I hope the authors will improve on these.

First of all, I found the explanation about the interpolation of the different data not very clear (P9.L251-256). I had to read the paragraph a couple of times to understand what you do. It seems to me that this is however a key explanation, as you are comparing spatial scales, and this explanation deserves therefore more space and elaboration. Please take the reader by the hand and try to explain stepwise what you do and elaborate a bit. State clearly what the original resolution is, what the interpolated resolution is and how you then interpolate again to the land surface resolution. It is especially important to emphasize these different steps, because I had initially the impression that you could skip the interpolation of the WRF-model and interpolate directly to the land surface resolution. But, if I understood correctly in the end, this is not the case, correct?

The authors also draw the conclusions that the topography matters in some cases, and in some cases more the atmospheric resolution. And yes, I agree with it that this can be seen qualitatively from the plots, but I believe the authors can easily make these statements much stronger. For example, the elevation data could be correlated with the errors, which allows for statistical proof that topography matters here. This can as well be done for north- or south-facing cells, as well as slopes. The errors can even be correlated with the atmospheric resolution, by calculating for the higher resolution land surface cells the distances to the center of the atmospheric cell.

In addition, I am also wondering if the Parflow model shouldn't be calibrated. As far as I understood, the authors change just the Manning coefficient manually. My first question here is, why the Manning coefficient? Parflow has many more parameters that can and maybe should be adjusted. And why is it done manually and what are the criteria for a good match? I believe it would be good to have a proper calibration, as comparing to a benchmark model that could be, in the most extreme case, be really bad does not make sense, and "errors" that are found could actually be improvements. The authors show some timeseries in Figure 7 of soil moisture, but how well does the model reproduce observed discharge? At least, if the authors stick with the uncalibrated model, please report how the performance of this model is with some metrics, this could give the reader some trust in the model.

The authors also never mention the model periods. Several times the year 2009 comes forward, but is this also the year you use for the model simulations? Generally, running a hydrological model for just one year seems really short to me and the outcomes could strongly depend on the conditions for that specific year (e.g. how many snow days were there that year?), also in combination with the initial conditions. If so, please report this, and add this to the limitations in the discussion. I believe it would

also be good if the authors test the robustness of their findings by changing the initial states of the model.

I believe the manuscript has potential. I hope the authors find my comments useful and I look forward to an improved version of the manuscript.

**Minor Comments**

P2.L58. Hillslope-scape → hillslope-scale?
P3.L83 deficits. and → remove period
P5.L143 (Kollet and Maxwell, 2008) → Kollet and Maxwell (2008)
P6.L161.  Hydrometerological → hydrometeorological
P6.Eq1. Please be consistent, ME is BIAS in the figures.
P7.L204. Neglecting… snow layers). → What do you mean here? Can you clarify?
P7.L204-207. The mass.. as follows. → This sentence seems a bit off, can you clarify and rephrase?
P8.L210. In snowpack → in the snowpack?
P8.L224. Equivalent are located → equivalent that are located?
P8.L225.  Was the data just retrieved in 2009 or are the observational data covering the year 2009?
P8.L234. Please add the reference of Van Genuchten (1980)
P8.L234. were retrieved from Leij (1996) and Simmers (2005) while land cover datasets, → were retrieved from Leij (1996) and Simmers (2005), while land cover datasets
P8.L237. by CLM model → by the CLM model
P9.L242. What do you mean with the modeling requirements here?
P9.L258. What do you mean with dampening the domain?
P9.L265. September 2008. → Do the model runs then start in October 2008?
P9.L277. During 2009 → during the 2009
P10.L292-294. In terms...grid resolution. → Please be specific how to see this. Is this the first row in Fig.8?
P10.produce of high → produce high?
P11.300-303. In these cases…lower elevated areas.-→ What do you mean? Please rephrase and clarify.
P11.L305. Tables 2 and 3 → tables 3 and 4?
P11.308. Specially for SWE, when degrading the hydrologic scale. → This doesn't seem to be true in all cases, I see it for A2, but A3H2-A3H3 goes from 42.03 to 41.61 mm, and A4H1-A4H2-A4H3 decrease from 60.64 to 57.26 and 56.53 mm. Seems that degrading the hydrologic scale matters more for a higher resolution forcing.
P11.L317. ( 11) → (Figure 11)
P12.L333.However, as decreasing –> However, when decreasing?

Fig.7. Is the unit mm/s for rainfall correct?
Fig.8.,9, 10, 11. Please add in the caption what each subfigure is. Why is this year 2014 and not 2009?

Are the data and modeling results publicly available?

---

## Short Comment (SC1) · 4 Dec 2020

Examining cross-scale influences of forcing resolutions in a hillslope-resolving, integrated hydrologic model Miguel A. Aguayo1, Alejandro N. Flores2, James P. McNamara2, Hans-Peter Marshall2, and Jodi Mead2

Despite the efforts made, this paper's objectives are vague. The overarching goal of this study is to assess the impact of the atmospheric forcing scales on hydrologic state variables such as snow water equivalent and soil moisture to potentially establish requirements for hydrologic modelling and observation system design". To make strengthening this paper, I'd recommend that the authors make their objectives straight-
forward and narrow into specific objectives and hypotheses. In this paper, the important words to set up the objectives have been missed namely: what, where, when, why etc.?

General comments and recommendation In this study, the authors designed numerical modelling experiments to analyze forcing resolutions in a hillslope-resolving, integrated hydrologic model. They used ParFlow-CLM to simulate the soil water and snow melting to assess the impact of the atmospheric forcing scales on hydrologic state variables. They modelled and compared the experiments at 30m, 90m and 250m of Hydraulic Resolution and 1 km, 3 km and 9 km of Atmospheric Forcing Resolution. First, the manuscript needs further editorial work to improve the paragraph structure and some vague expressions. Many sentences in this article are long with complicated structures and parenthesis in the middle, which makes it extremely difficult for the reader to follow and understand. I hope the authors make a change to improve the manuscript's clarity and this improvement in clarity will lead to better uptake and use of the ideas in the paper. Also, some references have not been provided, and the current references are not used consistently. Besides, there are many examples that are written in the parenthesis that could be brought out of the parenthesis. The comments are detailed in the technical issues. Secondly, some verification of the data is needed. Meaning, the finite element methods should be clarified. For example, if the incremental Jacobian method is the basis of the calculation, it should be mentioned and parametrized like shown in Figure 3. Also, it is better to parameterize the elements for the initial conditions of forcing data generated by the WRF model. To verify the method, calibration of the parameters and obtaining a more compatible diagram for Soil moisture, figure 7, can consolidate and support your results. Finally, the atmospheric model was not discussed in the Methodology, and references to other articles are used to prove the validation. Even though this area is well known and well used, you need to mention how you used the equations in your study. Lastly, I think you need to clear that this paper is methodly paper or a processing paper. If this is a processing paper, more results should be presented in the abstract instead of focusing on the methodology. Also, the abstract is too long, with about 480 words. A regular abstract usually includes

250 words. It would be best to rewrite the abstract and provide a concise of your achievements in this study. Technically I have some comments that might be useful for improving this study and manuscript: P1, L6, "This class of models is" is correct P1, L7-9, You have Used too many "and." You can use the comma instead of 'and.' P1, L11-14, The used sentence is long and heavy, and it could be broken into two sentences to be read easily. For instance, "Here we investigate the hydrologic impact of discrepancies between distributed meteorological forcing data. We can do this by exhibiting a range of spatial scales consistent with a variety of numerical weather prediction models when used to force an integrated hydrologic model associated with a corresponding range of spatial resolutions characteristic of distributed hydrologic modelling." P2, L32-34, Rewrite the sentence in an active voice, move the reference to the end (or be consistent in using of references) and correct Havana et al. (phi is not in the characters) P2, L35-37, The sentence is not clear for the calibration of parameters. P2, L41, You do not need parenthesis for each example. P2, L43, Use 'has' instead of 'have.' P2, L44-45, Provide a reference for the statements. Also, you do not need parenthesis for each example. P2, L49, Provide a reference for the paragraph. P2, L56, Use 'difference' instead of 'different.' P2, L58, Eliminate 'to' in "captured to within" in the beginning line and provide a paragraph reference. P3, L67, 'There are a few studies that examine' seems a better start instead of 'Even fewer of these studies examine.' P3, L75, Use estimation instead of estimate. P3, L76, Use investigates instead of 'investigate.' P3, L75-77, This sentence is not correct and necessary. There might be other studies that you have neglected. P3, L76-78, The sentence is too long and please rewrite it again in a simple way. P3, L82, Add 'and' after the word terrain. P3, L83, 'and also' is wrong. 'Also' is enough. P3, L89, 'that through a series' you can omit that. P3, L92, Which allows P3-4, In the last two paragraphs, the goals are divided into two parts, and it is better to be continuous. For this, you can shift the paper organization (L96,97) before the goals at the first of the paragraph. P5, L126, You do not need parenthesis for Prein et al., and you can rewrite the sentence for having a better look P5, L130, Have shown is correct. P5, L145,

[Figure]

You do not need the parenthesis for the examples. P5, L152-153, You do not need to write, i.e. for the different conditions. P6, L165, Suggested value is correct P6, L177, "Decided arbitrary" does not seem proper in this sentence. P7, L188, "hydrologic variables of importance" does not seem proper in this sentence. P7, L191, Please provide a reference for equation 3. P7, L195, Where is equation 3.2 P7, L198, Please rewrite the sentence. For example, "respectively. and $\mu$ represent ..." P7, L200, What does "It" refers to? P7, L207, Please provide a reference for the equation. P8, L212, Please provide a reference for the equation. P8, L214, In the sentence 'volume of the water l and ice i.' you can delete i and l or put them into parenthesis. P9, L260, I think it is better to define the initial condition for any finite element method based on the parameters you have specified in the last part. In other words, write the equation with initial conditions in a mathematical way for each initial condition. It can make the method more precise. P10, L277, Wrong spelling of discrepancy. P10, L277-281, This is not the article's purpose, but this is where you verify your simulation and show your method is compatible with the experimental data. The time differences are acceptable in figure 7, but I think the calibrations could help your data competence. P10, L296, Omit "of" and "in." "Produce of" and "across in." P11, L299, Keep the units of the horizontal axis in figures 9, 10, 11 consistent (mm or Cm) P11, L302-303, This sentence seems wrong. "The comparison shows topography? " Also, you used too many "on the other hand" expressions that are not necessary. P11, L317, What is (11) in the line? P12, L345, What is "It is" in the middle of the line? P12-13, L356-371, This part is discussion. You can use others' expressions and ideas and references them, but writing two paragraphs in the literature review style is not correct. What you have written is a part of the literature review. P13, L372, I think "Lie in" is not a proper verb in this sentence. P12, L376, I think "open avenues" is not a fair statement grammatically nor professionally. P13, L380, Please put a dot at the end of the sentence.

Please also note the supplement to this comment:
https://hess.copernicus.org/preprints/hess-2020-451/hess-2020-451-SC1-
supplement.pdf

---

## Referee Comment (RC2) · Anonymous Referee #2 · 19 Dec 2020

In the manuscript "Examining cross-scale influences of forcing resolutions in a hillslope-resolving, integrated hydrologic model" the authors present a numerical study of the impact of spatial scale of both forcing data and surface discretization on two distributed hydrological model states – soil moisture and snow water equivalent.

My broad concerns with this manuscript revolve around a) the limited details regarding the process representations used; and b) the limited mechanistic description of why the results are as they are. Despite what is clearly a lot of effort, the direct implications of the study are well known and not particularly novel. Certainly, the impact of atmospheric forcing resolution (Materia et al, 2010; Rasmussen, et al 2010; Bonekamp, et

al, 2018; Vionnet, et al. 2020), DEM resolution (Schlögl, et al 2016; Metcalfe, et al. 2015; Rasmussen, et al 2010), forcing errors (Raleigh, et al. 2015), and process scale (Klemeš, 1983) on both snowcover and hydrology are well known (to cite but a few examples). I will elaborate on these points below.

I have a major concern with this work viz a viz the (lack of) details regarding key-processes. The authors note that as the computational grid becomes more coarse, fine scale topography smooths out and thus impacts the surface energetics and subsequent SWE distributions. Certainly, this was to have been expected and especially the case with snow cover simulation in mountain terrain where smaller computational cells can better represent the heterogeneities in mass and energy (Dornes, et al, 2008; Schlögl, et al 2016; DeBeer, et al 2017; Vionnet, et al. 2020). However, there is seemingly no consideration of any of the small- to medium-scale (i.e., the 50 m to 250 m length scales considered) processes that might be affected by the scale perturbations, and no mechanistic discussion thereof to put the results into context. For example, there is no description of canopy interception of precipitation, nor if shortwave radiation to the slopes is cosine-corrected to take into account slope and aspect nor a discussion on shadowing, a key component in mountain terrain (Marsh, et al 2012). There is no discussion on the mechanical impact of terrain on wind fields, a key consideration in alpine terrain where overlapping terrain impacts makes simple spatial interpolation insufficient (Ryan, 1977). Specifically, wind accelerates over ridge crests and decelerates on the lee-slopes (Wood, et al, 2000). Near ridges, these high windspeeds can greatly increase sublimation (Mott et al, 2018) and cause blowing snow that further increase rates of snow sublimation. Mass lost to mid-winter sublimation can be as high as 10% to 30% (Mott et al, 2018), and is a critical component for evaluating total end of winter snow mass. The authors idly muse that "[. . .] might demonstrate that snow processes are not only dependent on orographic characteristics at that specific spatial scale, but also are highly dependent on the complex interaction between mass and energy inputs coming from the atmospheric processes and local topography." This is such a trivial, well-known statement I am uncertain why they hedge it with 'might'! Yet,

[Figure]

such a limited analysis and an obvious statement does not help the reader fully understand the impacts of the various numerical experiments. The interplay of topographic scale and NWP resolution on such key processes was missing from the analysis. It is not clear if these processes are implicitly treated, ignored, or not considered. The above noted processes are required for realistic snowmelt modelling and at the very least in such an analysis should be mentioned. This is to say nothing of the topographic impacts on the hydrology impacting soil moisture, however I hope my point is clear.

Another concern I have is with respect to the meteorological downscaling. In fact, I do not think it was done? As far as I could tell, there is no indication as to what geopotential or atmospheric height the forcing data were extracted from the WRF simulation. If a 'surface' prognostic variable was taken this can be problematic as, depending on the coarseness of the NWP resolution, the NWP terrain might be below the higher-resolution CLM topography. Thus, there may be cold biases, longwave biases, et cetera that are not considered (e.g., Bonekamp, et al, 2018). Further, this is a major complication with how windflow is treated. As noted above, high winds on crests (not even considering blowing snow) can lead to large sublimation losses, and differences in these should be visible between a 50 m to 250 m terrain model especially when forced with a 1km and a 9 km WRF simulation. However, the results and description lead me to believe that winds and meteorological forcing were simply assigned from a Xkm WRF simulation to the CLM terrain. I struggle to reconcile a decision not to downscale any meteorological forcing with running a 30 m hydrological model in alpine terrain.

A key detail and piece of analysis that is not mentioned is if the same amount of mass is input from the 1 km WRF simulations versus the 9 km simulations. Specifically, there is no reason to expect that cumulative seasonal precipitation is the same between the 1 km and 9 km WRF simulations (e.g., Bonekamp, et al, 2018; Rasmussen, et al 2010). Thus, the difference in results between the baseline and the various spatial scale perturbations is going to be a combination of small-scale process representation and differences in mass input. I realize scope with these types of numerical experiments can quickly get out of hand with "what if!" type questions. However, I am disappointed there is no comparison of the downscaled (however it was done) WRF forcing data with observations. I think this would have helped gauge if a 9 km WRF simulation was even close to providing anything but large-scale synoptic conditions or that it may be totally unsuitable for any type of hydrological modelling. Does it provide a precipitation mass estimate that is remotely close to correct?

Lastly, I think a major short coming of the manuscript is a lack of statistical analysis of the errors. The authors postulate a north-south distribution of errors, however there is no rigour in this statement to truly quantify this. Nor the elevation dependence on RMSE. The author's state "requires accurate and timely knowledge of runoff generated by snowmelt for water management" however no attention is given to melt timing, the impact of scale on this, nor if the melt timing produced by any of the combinations is close to observed melt timing.

As a note on the distributed RMSE, I wonder if the authors considered other ways of expressing the error metrics? The RMSE plots (Figure 8,9) have the WRF resolution artifact which is rather distracting. Distributions of RMSE or perhaps something like the Wasserstein metric may help the multi-scale analysis.

Regrettably, in the current form, I cannot recommend this manuscript for publication.

Specific points follow:

Figure 1 needs a legend

L1 I found the abstract quite long and should be tightened up

L5 "many issues" – a bunch of effort is spent making it seem liked distributed models are highly uncertain (which is absolutely a fair concern) and yet they are used herein. I think this should be tightened up, especially in the abstract

L13 "discrepancies" word choice

[Figure]

L33 "those" = what?

L34 "prove performance" debateable, lots of simple models completely fail in cold region, alpine terrain. Either way, citation needed.

L35 "watershed processes. . .properties" such as?

L38 "Because these models require. . .impossible to validate" I think I understand what you're saying here but could be tightened up. By construction, any model is impossible to validate in an ungagued basin.

L39 "Inc, 2015" confirm Inc is right

L39 & L43 Are you using integrated and physically based synonymously?

L48 "often not representative" citation needed

L49 "radar-retrieved precipitation" can also note low elevation gauge bias and/or total lack of gauging in complex terrain.

L51 "these models" which models?

L54 "spatial scales" spatial and temporal?

L67 "fewer of these studies" which studies?

L75 "To our knowledge only one study. . ." I would suggest that is not correct. A few examples: Rasmussen, et al 2010; Ikeda, et al. 2010; Bonekamp, et al, 2018; Vionnet, et al. 2020

L93: "more integrated way" what does this mean? Process coupling? Process representation?

L98: Discussion should be lower-case d

L106: "facing" word choice

L125 How fine of a scale is 'fine' referring to?
L125 'Within this' what is this referring to?

L127 , missing after e.g.

L139 remove "also" after ParFlow

L143 Extra ( ) around Kollet citation

L144,145 double space after the e.g.,

L159 "forcing data" note what variables are being used

L161 "synthesizing" what does this mean in this context?

L165 I'm a bit surprised this was the only variable calibrated

L174 Personally I am not a fan of 'Nature run' or 'Truth'. Without any in situ observations we don't know what the 'truth' is (which is already a loaded term with models), and 'Nature run' makes me think 'naturalized flow'. I think 'reference' or 'baseline' would be more appropriate.

L188 "hydrological <state> variables"

L190 This entire section is missing variable units!

L212 How is net radiation calculated i.e., albedo parameterization? Key for how it interacts with your scale perturbations

L218 What z0 are you assuming for your turbulent heat fluxes? What stability parametrizations are being used??

L258 What does 'dampening the domain' mean in this context?

L260 "Afterwards", after what?

L270 "short-term simulation" what is the length of this?

L275-281 Reads like Results?

L317 double space in front of 11

L368 "Decreases in terrain resolution may lead..." potentially, but it can also lead to increased uncertainty due increased number of parameters and process scale considerations. In cold regions smaller scale tends to result in better process representation due to capture small scale heterogeneities in mass and energy, however the rainfall-runoff literature tends to find increased uncertainty! This is a very nuanced point with a lot of literature supporting multiple sides that, in my opinion, deserves more than this one liner.

References

Bonekamp, P. N. J., Collier, E. & Immerzeel, W. W. The impact of spatial resolution, landuse and spin-up time on resolving spatial precipitation patterns in the Himalayas The impact of spatial resolution, landuse and spin-up time on resolving spatial precipitation patterns in the Himalayas. J Hydrometeorol 19, 1565–1581 (2018).

DeBeer, C. M. & Pomeroy, J. W. Influence of snowpack and melt energy heterogeneity on snow cover depletion and snowmelt runoff simulation in a cold mountain environment. J Hydrol 553, 199–213 (2017).

Dornes, P., Pomeroy, J. W., Pietroniro, A. & Verseghy, D. L. Effects of Spatial Aggregation of Initial Conditions and Forcing Data on Modeling Snowmelt Using a Land Surface Scheme. Journal of Hydrometeorology 9, 789–803 (2008).

Ikeda, K. et al. Simulation of seasonal snowfall over Colorado. Atmos Res 97, 462–477 (2010).

Materia, S., Dirmeyer, P. A., Guo, Z., Alessandri, A. & Navarra, A. The Sensitivity of Simulated River Discharge to Land Surface Representation and Meteorological Forcings. J Hydrometeorol 11, 334–351 (2010).

Marsh, C. B., Pomeroy, J. W. & Spiteri, R. J. Implications of mountain shading on calculating energy for snowmelt using unstructured triangular meshes. Hydrol Process

26, 1767–1778 (2012).

Metcalfe, P., Beven, K. and Freer, J.: Dynamic TOPMODEL: A new implementation in R and its sensitivity to time and space steps, Environmental Modelling & Software, 72, 155 172, doi:10.1016/j.envsoft.2015.06.010, 2015.

Mizukami, N. et al. Hydrologic Implications of Different Large-Scale Meteorological Model Forcing Datasets in Mountainous Regions. J Hydrometeorol 15, 474–488 (2014).

Mott, R., Vionnet, V. & Grünewald, T. The Seasonal Snow Cover Dynamics: Review on Wind-Driven Coupling Processes. Frontiers Earth Sci 6, 197 (2018).

Klemeš, V.: Conceptualization and scale in hydrology, Journal of Hydrology, 65(1–3), 1 23, doi:10.1016/0022-1694(83)90208-1, 1983. Raleigh, M. S., Lundquist, J. D. and Clark, M. P.: Exploring the impact of forcing error characteristics on physically based snow simulations within a global sensitivity analysis framework, Hydrology and Earth System Sciences, 19(7), 3153–3179, doi:10.5194/hess-19-3153-2015, 2015.

Rasmussen, R. et al. High-Resolution Coupled Climate Runoff Simulations of Seasonal Snowfall over Colorado: A Process Study of Current and Warmer Climate. J Climate 24, 3015–3048 (2011).

Ryan, B. C. A Mathematical Model for Diagnosis and Prediction of Surface Winds in Mountainous Terrain. Journal of Applied Meteorology 16, 571 584 (1977).

Schlögl, S., Marty, C., Bavay, M. and Lehning, M.: Sensitivity of Alpine3D modeled snow cover to modifications in DEM resolution, station coverage and meteorological input quantities, Environmental Modelling & Software, 83, 387 396, doi:10.1016/j.envsoft.2016.02.017, 2016.

Vionnet, V., Fortin, V., Gaborit, E., Roy, G., Abrahamowicz, M., Gasset, N. and Pomeroy, J. W.: Assessing the factors governing the ability to predict late-spring flooding in cold-region mountain basins, Hydrol Earth Syst Sc, 24(4), 2141–2165,

doi:10.5194/hess-24-2141-2020, 2020.

Wood, N. Wind Flow Over Complex Terrain: A Historical Perspective and the Prospect for Large-Eddy Modelling. Bound-lay Meteorol 96, 11–32 (2000).

---

## Author Comment (AC1) · 3 Feb 2021

Regarding the first question of the referee, the real resolutions for atmospheric forcings are 9 km, 3 km, 1 km, and they were used as input for 250 m, 90 m and 30 m surface hydrologic model. In addition, we interpolated 1 km atmospheric forcing to 30 m and use that dataset as input for 30 m surface specifically to generate the reference, and also we use this interpolated forcing data to run the hydrologic model to a 90 m and 250 m land surface resolution, respectively.

An example of how we performed the experiments is when using 30 m atmospheric resolution applied to a 90 m land surface resolution. Here, we subdivide the surface

cell into 30 m pieces by using nearest-neighbor resampling method in order to maintain the values of the 90 m land surface cells intact as shown in Fig 1. To clarify this point, we will provide a more elaborated paragraph to explain the interpolation process in both atmospheric and surface resolutions.

Regarding the referee's suggestion in which errors can even be correlated with both elevation and the atmospheric resolution, by calculating the distances for the higher resolution land surface cells to the center of the atmospheric cell, we will incorporate both correlation analyses to prove statistically that topography and atmospheric resolution matters.

On the model calibration comment, we agree Parflow has many more parameters that can be adjusted. However, a multiple parameter calibration for Parflow is still a computational challenge. The results should be seen as numerical experiments designed to assess scale issues in hydrologic modeling, and it is not intended to match observations to assess model performance with respect to observational data.

As the referee mentions, running a hydrological model for just one year seems really short and the outcomes could strongly depend on the conditions for that specific year. We agree with it, however longer simulations are computationally expensive, and we have chosen that year randomly among others with similar climatic conditions that match a dry initial condition at the end of summer followed by normal observed rain and snowfall in the study area and that also could allow us to analyze the hydrologic state variables chosen for these experiments. To clarify this part of the manuscript, we will consider the limitations of models and experiments in the discussion. Also, in an updated version of the manuscript, we will include streamflow simulated data to assess the atmospheric scale influences based on the mentioned variable.

Atmospheric forcing
Resolution (30 m)

30 m

Land Surface resolution
(90 m)

90 m

Land Surface resolution (90 m)
subdivided according to the
atmospheric forcing cells (30 m)
using Nearest-Neighbor
interpolation

**Fig. 1.** Example of 30 m atmospheric resolution applied to a 90 m land surface resolution.

---

## Author Comment (AC2) · 3 Feb 2021

Regarding the first comment and recommendation made, we will take the recommendation of making the objectives straightforward and try to narrow them into specific objectives. Also, we will improve paragraphs structure and some vague expressions by taking out parenthesis in the middle in some identified long sentences and rewriting such long sentences into short ones.

Regarding the second comment, Parflow uses Newton–Krylov-multigrid solvers, and it is fully explained in Jones, J. E., & Woodward, C. S. (2001) as cited in the manuscript. However, we will clarify the methods and the basis of the model calculation as sug-

gested.

Also, we agree that it is important to mention the parameterization schemes used in WRF that generated the initial conditions for this set of experiments and also incorporate a description of how the model equations are used in this study. We will incorporate the above mentioned in an updated version of the manuscript.

Regarding verify the method, calibration of the parameters, and obtaining a more compatible diagram for Soil moisture, we agree Parflow model needs more calibration, and as it is explained in responses to referee #1, it has many more parameters that can be adjusted. However, a multiple parameter calibration for Parflow is still a computational challenge. The results for this work should be seen as numerical experiments designed to assess scale issues in hydrologic modeling, and it is not intended to match observations to assess model performance regarding observational data.

For the last comment made, we will highlight that this is a methodology paper, we will consider all the minor comments suggested improving the manuscript, and we will definitely reduce the abstract to the usual size.

---

## Author Comment (AC3) · 3 Feb 2021

This response is regarding the broad concerns of referee 2 which revolve around a) the limited details regarding the process representations used; and b) the limited mechanistic description of why the results are as they are.

For the first concern (a) and question associated, what this manuscript intent to show is how combinations of scales can introduce systematic biases within integrated atmospheric and hydrologic modeling through synthetic experiments and independent of how well-calibrated the models are. It is understood that many complex interactions and processes occur in mountain ridges, sublimation, mass loss, etc. However, most

of the LSM lack representation of snow blowing processes that affect, without a doubt, snow-mass in mountain topography and therefore energy and momentum transfer to the atmosphere. In this manuscript, we introduce a methodological framework that works under such model limitations and intends to show how scale "in modeling" might affect predictions using synthetic experiments mixing scales for some specific hydrologic variables like in the OSSEs.

For the second concern (b) and questions associated, we took the forcing from the surface (2 m and 10 m) as required by Parflow-CLM and this is what most modelers usually do. The meteorological downscaling was done physically up to 1km resolution since 3 nested domains were considered in WRF simulations (9,3,1 km). We agree that there are not considerations on cold biases, how windflow is treated, large sublimation losses in 30 m interpolations as the referee mentioned, however, as above mentioned this work was done only under integrated models' capabilities can reach. For this major concern, we will incorporate the caveats of using integrated hydrologic models in the discussion since typically input forcing data (interpolated or not) are often used by modelers, not considering the key processes mentioned by the referee. Regarding reconcile the decision not to downscale any meteorological forcing with running a 30 m hydrological model in alpine terrain, we thought that it would be ideal, however, running WRF at 30 m spatial resolution with hourly outputs for a water year is computational and data storage costly, and it was out of our computational capabilities.

Regarding the major shortcoming of the manuscript, we will give more attention to snowmelt timing, and also incorporating other ways to express the error metrics such as the mentioned Wasserstein metric or others to get a less distracting multi-scale analysis.

In summary, to improve this manuscript we will include the following points based on all suggestions made by the referee, such as:

- Include CLM snow modeling limitations in the discussion.

- Incorporate the interplay of topographic scale and NWP resolution key processes

- Provide results and analysis using observed precipitation data from DCEW to see if WRF precipitation mass estimate is close to observation.

- We will consider the ways of expressing the error metrics such as mentioned Wasserstein metric or others that we will in the next manuscript phase.

- We will ditch all the OSSE terminology that might be potentially confusing. Since we didn't really do an OSSE where we assimilated data, we will adopt some other language like "synthetic experiments" in which we use the ParFlow model as a numerical laboratory where we can examine process interactions under controlled settings.

- Along with every point mentioned above, we will fix all the minor comments.